# Key needs, quality performance indicators and outcomes for patients with atrial fibrillation and multimorbidity: The AFFIRMO study

Donato Giuseppe Leo [1,2], Caterina Trevisan[3,4,5], Adele Ravelli[3], Trudie C. A. Lobban[6], Deirdre A. Lane [1,2,7] *, on behalf of the AFFIRMO Study investigators[¶]

1 Liverpool Centre for Cardiovascular Sciences, University of Liverpool and Liverpool Heart and Chest Hospital, Liverpool, United Kingdom, 2 Department of Cardiovascular and Metabolic Medicine, Faculty of Health and Life Sciences, Institute of Life Course and Medical Sciences, University of Liverpool, Liverpool, United Kingdom, 3 Department of Medicine, University of Padua, Padua, Italy, 4 Department of Medical Sciences, University of Ferrara, Ferrara, Italy, 5 Aging Research Center, Department of Neurobiology, Care Sciences and Society (NVS), Karolinska Institutet-Stockholm University, Stockholm, Sweden, 6 Arrhythmia Alliance, Stratford upon Avon, Warwickshire, United Kingdom, 7 Department of Clinical Medicine, Aalborg University, Aalborg, Denmark

¶ Membership of the AFFIRMO Study investigators is listed in the S1 File.
* deirdre.lane@liverpool.ac.uk

**Data Availability Statement:** Data cannot be publicly shared for ethical reasons, as data may contain potentially identifying or sensitive patient

## Abstract

### Background

Patients with atrial fibrillation (AF) often have concomitant long-term conditions that negatively impact their quality of life and the clinical management they receive. The AFFIRMO study aimed to identify the needs, quality performance indicators (QPIs), and outcomes relevant to patients, caregivers and healthcare professionals (HCPs) to improve the care of patients with AF.

### Methods

An on-line survey to collect the key needs, QPIs, and outcomes relevant to patients with AF, their caregivers and HCPs, was distributed between May 2022 and January 2023 in five countries (UK, Italy, Denmark, Romania and Spain). Results from the on-line survey were discussed in a three-round Delphi process with international representatives of patients with AF, caregivers, and HCPs to determine the key needs, QPIs and outcomes for the management of patients with AF and multimorbidity.

### Results

659 patients (47.2% males, mean (SD) age 70.9 (10.2) years), 201 caregivers (26.9% males, mean (SD) age: 58.3 (SD 15.2) years), and 445 HCPs (57.8% males, mean (SD) age 47.4 (10.6) years) participated in the survey. An initial list of 27 needs, 9 QPIs, and 17 outcomes were identified. Eight patients, two caregivers, and 11 HCPs participated in the

information, but data sets can be made available upon request to the corresponding author (deirdre. lane@liverpool.ac.uk) or the Sponsor at sponsor@liverpool.ac.uk

**Funding:** The AFFIRMO project has received funding from the European Union's Horizon 2020 research and innovation programme under Grant agreement No 899871.

**Competing interests:** DGL declares no conflict of interests. AR declares no conflict of interests. CT declares no conflict of interests. TL declares no conflict of interests. DAL: received investigator-initiated quality improvement grants from Bristol-Myers Squibb (BMS) and Pfizer, has been a speaker for Bayer, Boehringer Ingelheim, and BMS/ Pfizer and has consulted for BMS and Boehringer Ingelheim, all outside the submitted work.

**Abbreviations:** ABC, Atrial Fibrillation Better Care; ACE, Altarum Consumer Engagement Measure; AF, Atrial Fibrillation; AFFIRMO Study, Atrial Fibrillation Integrated Approach in Frail, Multimorbid, and Polymedicated Older People Study; BCOS, Bakas Caregiving Outcome Scale; CHE-s, Caregivers Health Engagement Scale; COMET Initiative, Core Outcomes Measures in Effectiveness Trials Initiative; GRADE guidelines, Grading of Recommendations, Assessment, Development, and Evaluations guidelines; HCCQ, Healthcare Climate Questionnaire; HCPs, Healthcare Professionals; MARS-5, 5-item Medication Adherence Report Scale; PHE-s, Patient Health Engagement; QPIs, Quality of Performance Indicators.

Delphi process. Nineteen (70%) needs, 8 (89%) QPIs, and 13 (76%) outcomes reached "consensus in", and were included in the final list.

## Conclusions

The final key needs, QPIs and outcomes obtained from the Delphi process will inform the AFFIRMO clinical trial, which aims to test the iABC app which incorporates an empowerment toolbox for patients and their caregivers, providing information to improve patient engagement and empowerment to help improve the clinical and self-management of patients with AF in the context of multimorbidity.

## Introduction

Atrial fibrillation (AF) affects 1–2% of the European population [1], with its incidence increasing particularly among people aged 65 years and older [1]. In recent years, the management of AF has embraced a more integrated care approach in the form of the ABC (Atrial Fibrillation Better Care) pathway [2–4], however there is still a large discrepancy in the management of patients within and between countries [4, 5].

Multimorbidity, defined as the concomitant presence of two or more chronic health conditions [6, 7], is very common in patients with AF, who often report a higher multimorbidity rate compared to the non-AF population [8, 9]. The most common comorbidities among patients with AF are hypertension, heart failure and chronic kidney disease, all of which are associated with a higher risk of hospitalisation, and all-cause mortality [9]. The burden of multimorbidity worsens with ageing, negatively affecting the patients' quality of life, and increasing the complexity of their clinical management [6, 7].

Ignoring the heterogeneous spectrum of disease combinations typical of multimorbidity, the majority of the European healthcare systems still adopt a single-disease approach, which reduces the likelihood of a holistic approach, taking into account the overall health of the patient [10]. A more patient-centred approach is advocated in order to address the needs of patients with AF in the context of multimorbidity [11, 12], also considering key factors such as increased patient education and health literacy [13], and assessment of quality performance indicators (QPIs) of care [14]. Moreover, it is crucial empowering patients to be more involved in the management of their health and to become partners with the healthcare team in the decision-making process concerning their care [15, 16].

The Atrial Fibrillation Integrated Approach in Frail, Multimorbid, and Polymedicated Older People (AFFIRMO) project [17] aims to improve the management of AF in the context of multimorbidity, with the focus on a holistic approach to optimise clinical management of older patients with AF, considering the multifaceted aspects of individuals' health, including multimorbidity, polypharmacy, personal preferences, and social context. The AFFIRMO substudy reported here aimed to identify the key needs, QPIs, and outcomes relevant to patients with AF and multimorbidity, their caregivers, and healthcare professionals (HCP) involved in their clinical management. This aim was to understand the key aspects of care that need to be improved in the management of these patients and to inform the clinical trial of the AFFIRMO study, where an empowerment toolbox will be provided to patients and caregivers, enabling them to receive information about their health conditions.

## Methods

The identification of the key needs, QPIs and outcomes consisted of two phases (**Fig 1**): (i) an international on-line survey open to patients, their caregivers, and HCPs distributed in five countries (UK, Italy, Spain, Denmark, Romania) participating in the AFFIRMO study [17], and (ii) a Delphi process with international patients, caregivers, and HCP representatives.

Ethical approval from local and national authorities was sought and granted in the UK (REC 21/YH/0307), Italy (015534, ref. 5308/AO/22), Spain (2021-12-15-HCUVA), and Romania (2SNI/13.01.2022). Consultation with the Danish Research Ethics committee confirmed no need for ethical approval for this type of study in Denmark.

### Phase 1: On-line survey

An online survey platform was developed by AdvicePharma to host the international survey, and the content of the patient, caregiver and HCP surveys (**S2-S4 Tables in S1 File**, respectively) were developed by a multidisciplinary project team (DAL, CT, DGL, TL and GG) composed of psychologists, geriatrician, and an AF patients' representative. Validated questionnaires were included to assess patients quality of life (EQ-5D-3L [18]), perception of healthcare received (Healthcare Climate Questionnaire–HCCQ [19]), engagement in the healthcare process (Altarum Consumer Engagement Measure–ACE [20]; and the Patient Health Engagement Scale–PHE-s [21]), frailty (FRAIL questionnaire [22]), and medication adherence (5-item Medication Adherence Report Scale–MARS-5 [23]). In the caregiver survey, quality of life was assessed using the ED-5D-3L (ref), engagement in the healthcare process with the Caregivers Health Engagement Scale (CHE-s [24]), and level of life changes by the Bakas Caregiving Outcome Scale (BCOS) [25]).

In the HCPs survey, data related to demographics (age, sex, racial/ethnic origin), job role and related details (e.g. medical specialty; primary, secondary or tertiary care centre), and country of residence were collected. Additionally, the survey also aimed to seek HCPs view on

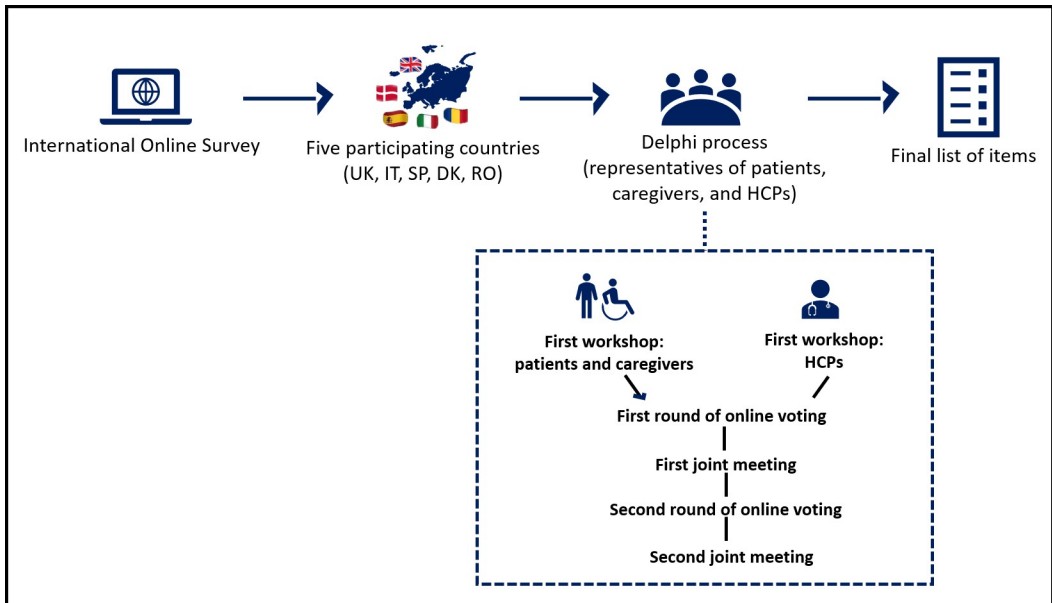

**Fig 1. Flow-chart of the study process.** UK = United Kingdom, IT = Italy, SP = Spain, DK = Denmark, RO = Romania, HCPs = Healthcare Professionals, QPIs = Quality of Performance Indicators.

the importance that they attribute to patient's engagement in the care management. The online survey was divided by groups: (i) patients, (ii) caregivers, and (iii) HCPs, each asking socio-demographic information (e.g., age, sex, level of education) and reporting specific self-reported questionnaires to identify key needs, QPIs, and key outcomes of treatment for the intended category. A list of the questionnaires used and the template of the survey can be found in Appendix 1. All three surveys were developed in English and then translated into Italian, Spanish, Danish and Romanian. Validated questionnaires that did not have a translation for the countries included in the online survey were translated by a professional company, and the back-translation was confirmed by the relevant country leads.

The survey was open to patients with AF and one or more concomitant chronic health conditions (multimorbidity), and their caregivers (dyads and non-dyads), and to HCPs managing patients with AF and one or more concomitant chronic health conditions. Exclusion criteria were as follows: (i) inability to consent, (ii) moderate or severe cognitive impairment (e.g. dementia), (iii) presence of moderate or severe conditions that may prevent completion of the survey, (iv) inability to complete the survey online, and (v) unwilling to participate in the study. Patients and caregivers were invited to participate in the survey via advertisement on a patient organisation website (Atrial Fibrillation Association, AFA), or by HCPs to patients attending clinical appointments in participating hospitals. HCPs were invited to participate by e-mail via professional networks. The first page of the online survey asked the participant to provide consent; this was mandatory prior to accessing the main survey. The online survey was distributed between 31 May 2022 and 31 January 2023 in the UK, Italy, Spain, Denmark, and Romania.

## Phase 2: Delphi process

The needs, QPIs, and outcomes identified from the survey (Phase 1) were discussed during a Delphi process [26] that involved three online meetings of approximately two hours each (via Zoom), and two rounds of independent off-line voting using the JISC survey platform. The inclusion criteria for the Delphi process were the same as the on-line survey, with the addition of being able to speak English, as a common language was required for the meetings. Participants were asked to sign and return a consent form via email. Participation required attendance at all the meetings and completion of all rounds of voting. The Delphi process ran between February and April 2023. The initial meeting was held separately for patients/caregivers and for HCPs. The patient-caregiver meeting was chaired by a patient advocate and the HCP meeting by a geriatrician, independent from the AFFIRMO project. The Chairs did not participate in the voting. A list of Delphi panel members is available in the S1 File.

After these meetings, the link to access the first round of voting was sent by email to all participants, with one week to respond. Participants were asked to score each item on a scale from 1 to 9, where 1–3 was defined as "not important at all", 4–6 as "important but not essential", and 7–9 as "extremely important". Only items scored as "extremely important" (7–9 score) by the 80% or more of the participants were included in the final list. Items scored as "not important at all" (1–3 score) by 80% or more of the participants were moved to the "excluded list" (**S1 Table in S1 File**). Items where a majority decision for inclusion/exclusion (those that did not reach ≥80%) was not reached, were carried over to the next online meeting for discussion.

The second online meeting was held jointly for all participants. The results from the first round of voting were presented (included, excluded and undecided (those scored 4–6)). Following this meeting, participants were sent the link to access the second round of voting by email. The final joint meeting summarised the results of the second round of voting. Needs, QPIs and outcomes that still did not reach consensus, were scored anonymously during the

meeting using VoxVote (VoxVote, The Netherlands). The final list of key needs, QPIs and outcomes was then presented to participants.

## Data analysis

Results of the online survey were reported using descriptive statistics (i.e. mean and standard deviation [SD] or median [interquartile range] for continuous variables normally or non-normally distributed, respectively; and counts/percentages for categorical data. Consensus on the inclusion/exclusion of the items presented during the Delphi process was defined in line with the GRADE guidelines [27] (details listed above).

# Results

## Online survey

A total of 1,305 participants completed the online survey. Of these, 659 (50.5%) were patients with a mean (SD) age of 70.9 (10.2) years and 348 (52.8%) female; 201 (%) were caregivers, mean (SD) age 58.3 (15.2) years and 147 (73.1%) female; and 445 (%) were HCPs, mean (SD) age 47.4 (10.6) and 257 (57.2%) female. More than half of the patients were from the UK (n = 358, 54.3%). Caregivers were mainly recruited from Spain, Romania and Italy, and HCPs were mainly recruited from the UK, Italy, Spain, and Romania. The distribution for each participant group by country is shown in **S1 Fig in S1 File**.

**Table 1** presents the characteristics of the patients who completed the online survey. Level of education among patients varied, with degree level or above being the most common (42.5%). Most patients were currently retired (78.5%), married/had a partner (68%), and lived at home with family without assistance (63.6%). When assistance was required, it was mainly informal (92.7%). Most patients reported having more than two comorbidities (45%), with 10% having more than five comorbidities. The most common comorbidities reported were other cardiovascular diseases (60.5%), hypertension (59%), and osteoarthritis (25%).

**Table 2** presents the characteristics of the caregivers who participated. Most caregivers reported spending less than 6h/day in caring activities (50.2%). Most were informal caregivers (91.5%), with more than five years as a caregiver (44.8%), mainly assisting a parent (36.8%), spouse/partner (27.9%) or other relative (25.4%). Less than half (46.8%) lived with the assisted person. The person they cared for had two or more comorbidities, predominantly cardiovascular conditions. Most assisted persons were taking more than five medications (70.6%), were able to walk independently (60.7%), but required assistance with some activities of daily living.

**Table 3** presents the characteristics of the HCPs that completed the online survey. Most HCP respondents were medical doctors (73.5%), either cardiologists (45.2%) or geriatricians (31.7%). Years of practice varied from less than five years (23.6%) to more than 30 years (20.9%). Most worked in secondary (36%) or tertiary care (38.7%) and managed two to five patients with AF per week, and the most common comorbidities were cardiovascular diseases (92.6%).

The list of needs, QPIs and key outcomes identified by the online survey (n = 53) are shown in **S1 Table in S1 File**.

## Delphi process

A total of 21 participants joined the Delphi panel: 8 patients, 2 caregivers and 11 HCPs. More than half of the participants were men (n = 11, 52%). Due to the requirement of being able to speak English, all patients and caregivers were from the UK (n = 10, 48%). Among the HCPs, there were three participants each from Italy and Denmark, two each from Spain and

**Table 1. Characteristics of the patients who participated in the international on-line survey.**

| Mean (SD), n (%) | Patient group (n = 659) |
|---|---|
| Age (years) | 70.9 (10.2) |
| Women | 348 (52.8) |
| Ethnicity | |
| White | 645 (97.9) |
| Hispanic or Latino | 6 (1.1) |
| Country | |
| UK | 358 (54.3) |
| Italy | 84 (12.7) |
| Spain | 122 (18.5) |
| Romania | 92 (14.0) |
| Denmark | 3 (0.5) |
| Level of Education | |
| None | 17 (2.7) |
| Primary | 77 (12.0) |
| Secondary | 51 (7.7) |
| High School* | 158 (24.0) |
| Apprenticeship/Professional training/vocational training* | 57 (8.6) |
| Degree level or above | 280 (42.5) |
| Other/prefer not to say | 19 (2.9) |
| Employment status | |
| Employed | 102 (15.5) |
| Unemployed | 20 (3.0) |
| Retired | 517 (78.5) |
| Disability Allowance | 20 (3.0) |
| Marital status | |
| Single/Never married | 39 (5.9) |
| Married/Partnered | 448 (68.0) |
| Widowed | 111 (16.8) |
| Separated/Divorced | 61 (9.3) |
| Living arrangements | |
| Living at home alone with no assistance | 168 (25.5) |
| Living at home with family with no assistance | 419 (63.6) |
| Living at home with part-time assistance | 50 (7.6) |
| Living at home with full-time assistance | 19 (2.9) |
| Living in long-term care facilities | 3 (0.5) |
| If assistance is required, the caregiver is | |
| Formal | 48 (7.3) |
| Informal | 611 (92.7) |
| Smoking status | |
| Current | 33 (5.0) |
| Former | 281 (42.6) |
| Never | 345 (52.4) |
| Comorbidities (n, %) | |
| Hypertension | 389 (59.0) |
| Cardiovascular disease | 399 (60.5) |
| Diabetes mellitus | 108 (16.4) |
| Thyroid disease | 108 (16.4) |

(*Continued*)

**Table 1.** (Continued)

| Mean (SD), n (%) | Patient group (n = 659) |
|---|---|
| Chronic obstructive pulmonary disease | 41 (6.2) |
| Gastrointestinal diseases | 128 (19.4) |
| Chronic liver disease | 19 (2.9) |
| Kidney disease | 62 (9.4) |
| Previous stroke | 57 (8.6) |
| Parkinson's disease | 8 (1.2) |
| Multiple sclerosis | 3 (0.5) |
| Dementia | 2 (0.3) |
| Cognitive decline | 46 (7.0) |
| Osteoarthritis | 165 (25.0) |
| Osteoporosis/previous hip fracture | 51 (7.7) |
| Rheumatoid arthritis | 29 (4.4) |
| Chronic pain | 77 (11.7) |
| Vision problems | 119 (18.1) |
| Hearing problems | 105 (15.9) |
| Cancer | 40 (6.1) |
| Other | 182 (27.6) |
| Number of comorbidities | |
| None | 27 (4.0) |
| 1–2 comorbidities | 270 (41.0) |
| 3–5 comorbidities | 297 (45.0) |
| >5 comorbidities | 66 (10.0) |
| Hospital visits per year, median (IQR) | 0.0 (0.0–1.0) |

IQR, interquartile range; SD, standard deviation

Romania, and one from Australia. Of this group, three were nurses and eight were physicians, with four specialising in geriatrics and/or internal medicine and four in cardiology.

The key needs, QPIs and key outcomes identified by the online survey (**S1 Table in S1 File**) were presented to the Delphi panel at the initial meetings. Forty items reached 'consensus in' during the process and were included in the final list. These were grouped into 19 key needs, eight QPIs and 13 key outcomes (**Table 4 and Fig 2**).

## Discussion

In our study, we aimed to identify the key needs, QPIs, and outcomes relevant to multimorbid patients with AF, their caregivers, and the HCPs that manage their health, collecting data from an international online survey and reaching consensus through a Delphi panel with international stakeholders.

The online survey identified a preliminary list of 53 items, which were reduced to 40 items during the Delphi process and were included in the final list of key needs, quality performance indicators and outcomes. Of these, 19 key needs were identified including stroke prevention, symptom control, reducing adverse events/medication side effects, and improving patient education and HCP-patient communication. Eight QPIs were identified including appropriate treatment and medications relating to stroke prevention and to reduce adverse events (bleeding and cardiovascular events) and medication reviews to ensure optimal management and to reduce medication side effects, and provision of appropriate education tools for patients and

**Table 2. Characteristics of the caregivers who participated in the international on-line survey.**

| Mean (SD), n (%) | Caregiver group (n = 201) |
|---|---|
| Age (years) | 58.3 (15.2) |
| Women | 147 (73.1) |
| Ethnicity | |
| White | 199 (99.0) |
| Other | 2 (1.0) |
| Country | |
| UK | 6 (3.0) |
| Italy | 49 (24.4) |
| Spain | 66 (32.8) |
| Romania | 80 (39.8) |
| Denmark | 0 (0.0) |
| Level of Education | |
| None | 2 (1.0) |
| Primary | 23 (11.4) |
| Secondary | 22 (11.0) |
| High School | 40 (20.0) |
| Apprenticeship/Professional training/vocational training | 15 (7.5) |
| Degree level or above | 77 (35.0) |
| Other/prefer not to say | 22 (11.0) |
| Time spent caregiving | |
| Full-time | 43 (21.4) |
| Less than 6 h/day, daily | 57 (28.4) |
| Less than 6h/day, not daily | 101 (50.2) |
| Person assisted–if informal (n = 184) | |
| Spouse/partner | 56 (27.9) |
| Father | 31 (15.4) |
| Mother | 43 (21.4) |
| Any other relative | 51 (25.4) |
| A friend | 3 (1.5) |
| Living with assisted person | 94 (46.8) |
| Type of caregiver | |
| Formal | 17(8.5) |
| Informal | 184(91.5) |
| Years being caregiver | |
| ≤1 year | 50 (24.9) |
| 2–4 years | 61 (30.3) |
| ≥5 years | 90 (44.8) |
| Comorbidities of the assisted person | |
| High blood pressure | 124 (61.7) |
| Heart disease | 171 (85.1) |
| Diabetes | 66 (32.8) |
| Thyroid problems | 38 (18.9) |
| Chronic obstructive pulmonary disease | 24 (11.9) |
| Gastrointestinal diseases | 32 (15.9) |
| Chronic liver disease | 7 (3.5) |
| Kidney disease | 38 (18.9) |
| Previous stroke | 31 (15.4) |

(*Continued*)

**Table 2.** (Continued)

| Mean (SD), n (%) | Caregiver group (n = 201) |
|---|---|
| Parkinson's disease | 5 (2.5) |
| Multiple sclerosis | 1 (0.5) |
| Dementia | 11 (5.5) |
| Cognitive decline | 29 (14.4) |
| Osteoarthritis | 41 (20.4) |
| Osteoporosis/previous hip fracture | 22 (10.9) |
| Rheumatoid arthritis | 14 (7.0) |
| Chronic pain | 22 (10.9) |
| Vision problems | 41 (20.4) |
| Hearing problems | 42 (20.9) |
| Cancer | 22 (10.9) |
| Other | 27 (13.4) |
| Number of comorbidities of the assisted person | |
| None | 5 (2.5) |
| 1–2 comorbidities | 79 (39.3) |
| 3–5 comorbidities | 87 (43.3) |
| >5 comorbidities | 30 (15.0) |
| Number of medications taken by the assisted person | |
| None | 5 (2.5) |
| 1 to 2 | 10 (5.0) |
| 3 to 4 | 44 (21.9) |
| 5 or more | 142 (70.6) |
| Mobility level of the assisted person | |
| Can walk independently | 122 (60.7) |
| Walk with a cane/walking stick | 37 (18.4) |
| Walk with a walker/Zimmer-frame | 28 (13.9) |
| Moves around with a wheelchair | 5 (2.5) |
| Confined at home, mostly lying on the bed | 9 (4.5) |
| Activities that require assistance | |
| Eating | 37 (18.4) |
| Bathing | 81 (40.3) |
| Dressing | 45 (22.4) |
| Toileting | 35 (17.4) |
| Transferring | 141 (70.1) |

SD, standard deviation

caregivers. Thirteen key outcomes were identified including the ability to work, quality of life, preventing hospitalisation, stroke, and cardiovascular diseases, reducing adverse events/medication side effects, and symptom management.

The importance of a clear definition of key needs and outcomes in clinical research and practice has been previously highlighted [26, 28], and international initiatives such as the "Core Outcomes Measures in Effectiveness Trials" (COMET) Initiative [29] have established guidelines on how to construct core outcomes sets and how to involve key stakeholders (patients, caregivers and HCPs) in the process. However, until now there has been no comprehensive list of key needs, QPIs and outcomes for patients with AF and other concomitant long-term health conditions. The identification of a set of key items that are relevant for the

**Table 3. Characteristics of the healthcare professionals who participated in the international on-line survey.**

| Mean (SD), n (%) | Healthcare professional group (n = 445) |
|---|---|
| Age (years) | 47.4 (10.6) |
| Women | 257 (57.8) |
| Country | |
| UK | 72 (16.2) |
| Italy | 127 (28.5) |
| Spain | 120 (27.0) |
| Romania | 100 (22.5) |
| Denmark | 26 (5.8) |
| Occupation | |
| Medical doctor | 327 (73.5) |
| Nurse | 110 (24.7) |
| Other | 8 (1.8) |
| Specialty | |
| Cardiology | 201 (45.2) |
| GP | 17 (3.8) |
| Geriatrics/Elderly care | 141 (31.7) |
| Haematology | 8 (1.8) |
| Internal medicine | 38 (8.5) |
| Other | 40 (9.0) |
| Years of practice | |
| 0 to 5 | 105 (23.6) |
| 6 to 10 | 65 (14.6) |
| 11 to 20 | 105 (23.6) |
| 21 to 30 | 77 (17.3) |
| >30 | 93 (20.9) |
| Care sector | |
| Primary care | 113 (25.4) |
| Secondary care | 160 (36.0) |
| Tertiary care | 172 (38.7) |
| University Hospital | |
| Yes | 337 (75.7) |
| No | 108 (24.3) |
| Regularly working with patients with chronic condition(s) | |
| Yes | 430 (96.6) |
| Sometimes | 13 (2.9) |
| No | 2 (0.4) |
| Patients with AF managed per week | |
| 0 to 1 | 45 (10.1) |
| 2 to 5 | 231 (51.9) |
| 6 to 10 | 81 (18.2) |
| >10 | 88 (19.8) |
| Most frequently managed conditions | |
| Cardiovascular | 412 (92.6) |
| Diabetes | 172 (38.7) |
| Endocrinologic diseases | 6 (1.3) |
| Respiratory diseases | 136 (30.6) |
| Chronic liver diseases | 7 (1.6) |

*(Continued)*

**Table 3.** (Continued)

| Mean (SD), n (%) | Healthcare professional group (n = 445) |
|---|---|
| Gastrointestinal diseases | 13 (2.9) |
| Kidney diseases | 73 (16.4) |
| Cerebrovascular diseases | 71 (16.0) |
| Neurologic diseases | 19 (4.3) |
| Minor/major cognitive disorders | 86 (19.3) |
| Osteoarticular diseases | 25 (5.6) |
| Rheumatologic diseases | 9 (2.0) |
| Chronic pain | 16 (3.6) |
| Vision problems | 0 (0.0) |
| Hearing problems | 0 (0.0) |
| Other | 12 (2.7) |
| Most represented age group | |
| <60 years | 11 (2.5) |
| 60–70 years | 108 (24.3) |
| 71–80 years | 201 (45.2) |
| >80 years | 125 (28.1) |
| Assisted patients with AF and at least one other chronic condition | |
| 0 to 10% | 7 (1.6) |
| 11 to 30% | 28 (6.3) |
| 31 to 50% | 56 (12.6) |
| 51 to 80% | 141 (31.7) |
| >80% | 213 (47.9) |

AF, atrial fibrillation; SD, standard deviation

management of patients with AF and multimorbidity from the AFFIRMO study will help to inform and redirect the healthcare management of these patients toward a more holistic and unified approach [30], integrating the needs of patients and of the people that care for them into consideration, and focusing on QPIs and outcomes of importance to them. This list of items has informed the outcomes of the AFFIRMO clinical trial, which will hopefully help to improve the clinical management of patients with AF in the context of multimorbidity.

## Strengths and limitations

Our study collected data from a large international cohort of 1,305 patients, caregivers and HCPs from five European countries, identifying key needs, QPIs and outcomes of healthcare that are important to a vast range of stakeholders, also representing all regions of Europe (North, South and East Europe). The cohort was representative of the typical AF population in terms of age and comorbidities and included 57% females. The use of a consolidated methodology, such as the Delphi process, to reach consensus on the core items to report in the final list is a strength of our study. Involvement of patients' and caregiver representatives in the Delphi process, and also in the inclusion of a patient advocate as co-chair of the joint sessions, has mitigated the potential "medical-centred" discussion that could have arisen from a group of HCPs only. Patients and caregivers have been able to express their views, sometimes with constructive criticism on the medicalised approach they have experienced, thus improving the quality of the discussion during the Delphi process.

**Table 4. Final list of key needs, quality performance indicators and key outcomes reaching consensus.**

| Key Needs (n = 19) | Quality Performance Indicators (n = 8) | Key outcomes (n = 13) |
|---|---|---|
| • Stroke prevention<br>• Assessment and management of frailty<br>• Avoid hospitalization<br>• Balance benefit/risk ratio due to anticoagulant treatment<br>• Caregiver involvement in treatment decisions<br>• Co-morbidity management<br>• Control AF symptoms<br>• Control heart failure symptoms<br>• Control possible interactions between anticoagulant and other ongoing treatments*<br>• Improved communication between GP and other specialists<br>• Individual care plan<br>• Materials to explain conditions and treatment options<br>• Patient education<br>• Patient involvement in treatment decisions<br>• Reducing cardiovascular events (e.g., stroke, heart attack)<br>• Reducing medication side effects<br>• Reduction of major bleeding<br>• Social network/support<br>• Symptoms control (all symptoms) | • Appropriate prescription review to reduce drug interaction and medication side effects<br>• Appropriate prescription review to reduce the number of medications<br>• Appropriate resources (e.g., booklets, websites) to provide information on the management of the conditions and on medical recommendation to patients and their caregiver<br>• Appropriate stroke prevention/treatment<br>• Appropriate treatment to reduce the occurrence of cardiovascular events<br>• Appropriate treatment to reduce the risk of major bleeding<br>• Appropriate treatment to reduce/avoid hospitalization<br>• Reduction/alleviation of symptoms | • Ability to work<br>• Cognitive functioning<br>• Emotional functioning/ wellbeing<br>• Longevity/reducing mortality<br>• Maintaining independence<br>• Physical functioning<br>• Preventing heart failure<br>• Preventing major bleeding<br>• Preventing/reducing adverse treatment effects<br>• Preventing/reducing hospitalization<br>• Quality of Life<br>• Stroke prevention<br>• Symptoms reduction/ alleviation |

*described in Fig 2 as "Interaction of OAC (oral anti-coagulant) and other therapies".

Some limitations of the study are noteworthy. Patients and caregivers from Denmark and caregivers from the UK were under-represented in the online survey, mainly due to difficulties in engaging these groups in the respective countries. However, overall 50% and 15% of the

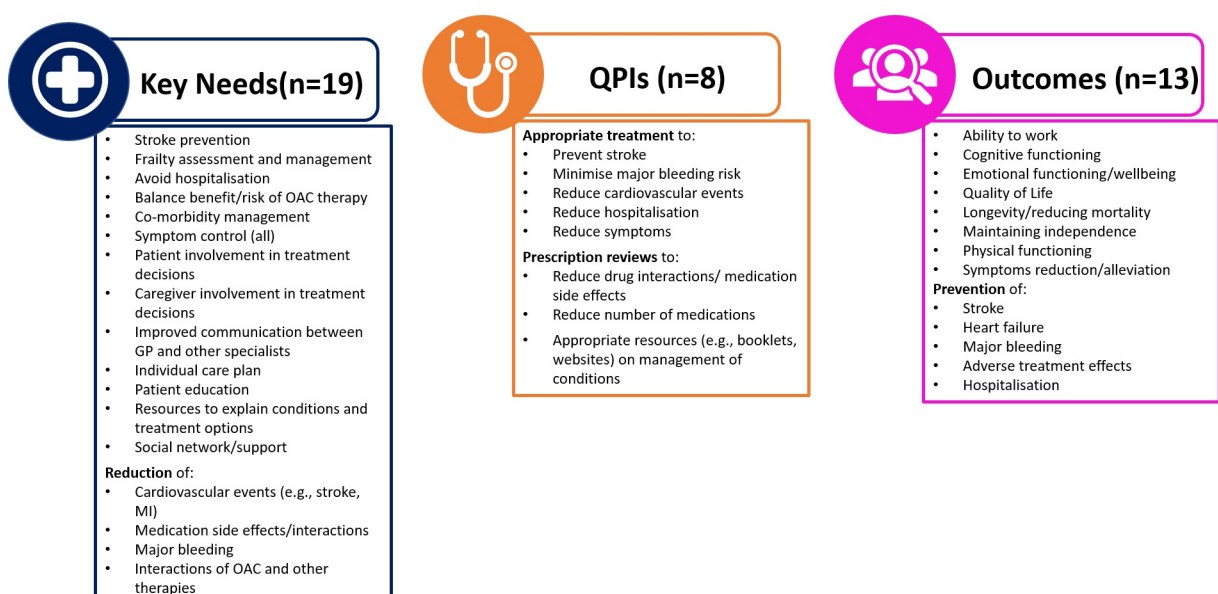

**Fig 2. Summary of the items that reached consensus in during the Delphi process.** OAC = oral anti-coagulant, QPIs = quality of performance indicators. Note: some rewording has been made in the Figure compared to the list in Table 4 for presentation purposes.

participants were patients and caregivers, respectively, and therefore, we believe the study has captured their 'voice' and there is no reason to believe that the opinions and insights of patients in Denmark and caregivers in the UK and Denmark would differ significantly from those captured in this study. The presence of only UK patient and caregiver participants in the Delphi process (due to the need of a common language for the discussion) is a limitation, which may have presented viewpoints related to one specific healthcare system, and may have influenced the key needs, QPIs, and outcomes selected. However, HCPs from five countries contributed to the Delphi process which ensured diversity in healthcare system perspectives. Further, the Delphi workshops discussed the needs, QPIs and outcomes identified by patients, caregivers and HCPs from five European healthcare systems via the on-line surveys, which minimises the possible UK-biased perspective of healthcare. Moreover, patients and caregivers who participated in the survey were mainly of White ethnicity which may reduce the generalisability of our findings.

## Conclusions

Forty items, divided into 19 key needs, eight QPIs, and 13 key outcomes, relevant to patients with AF and multimorbidity, their caregivers and HCPs were identified from an international survey and selected through a Delphi process. This list has informed the outcomes of the AFFIRMO clinical trial, which aims to direct the care of these patients to a more patient-centred approach.

## Supporting information

**S1 File. Supplementary materials.** List of AFFIRMO study investigators. List of Delphi Panel members. S1 Table. Full list of items identified from the survey that were scored in the Delphi process. S2 Table. Content of the online survey for patients. S3 Table. Content of the online survey for caregivers. S4 Table. Content of the online survey for healthcare professionals. (DOCX)

## Acknowledgments

The authors wish to thank Dr Asangaedem Akpan and Mrs Tracy Goodman for helping to co-chair the Delphi meetings, and for their general help and advice with this study. The authors would also like to thank all the participants of the Delphi meetings for their insightful comments (see S1 File for list).

## Author Contributions

**Conceptualization:** Deirdre A. Lane.

**Data curation:** Donato Giuseppe Leo, Caterina Trevisan, Adele Ravelli, Deirdre A. Lane.

**Formal analysis:** Donato Giuseppe Leo, Caterina Trevisan, Adele Ravelli, Deirdre A. Lane.

**Funding acquisition:** Deirdre A. Lane.

**Investigation:** Donato Giuseppe Leo, Caterina Trevisan, Adele Ravelli, Trudie C. A. Lobban, Deirdre A. Lane.

**Project administration:** Deirdre A. Lane.

**Supervision:** Deirdre A. Lane.

**Writing – original draft:** Donato Giuseppe Leo.

**Writing – review & editing:** Caterina Trevisan, Adele Ravelli, Trudie C. A. Lobban, Deirdre A. Lane.

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
