## [Decision Letter · Decision Letter 0]

28 Jun 2024

PONE-D-24-12600Key Needs, Quality Performance Indicators and Outcomes for patients with Atrial Fibrillation and Multimorbidity: The AFFIRMO Study.PLOS ONE

Dear Dr. Lane,

Thank you for submitting your manuscript to PLOS ONE. After careful consideration, we feel that your paper has merit however needs minor revisions and some editions/response to reviewers comments. Therefore, we invite you to submit a revised version of the manuscript that addresses the points raised during the review process.

We look forward to receiving your revised manuscript.

Kind regards,

Aviral Vij

Academic Editor

PLOS ONE

 [The AFFIRMO project has received funding from the European Union’s Horizon 2020 research and innovation programme under Grant agreement No 899871.].  

Additional Editor Comments (if provided):

Reviewers' comments:

Reviewer's Responses to Questions

**Comments to the Author**

1. Is the manuscript technically sound, and do the data support the conclusions?

Reviewer #1: Yes

Reviewer #2: Yes

Reviewer #3: Yes

2. Has the statistical analysis been performed appropriately and rigorously? 

Reviewer #1: N/A

Reviewer #2: Yes

Reviewer #3: Yes

3. Have the authors made all data underlying the findings in their manuscript fully available?

Reviewer #1: No

Reviewer #2: Yes

Reviewer #3: Yes

4. Is the manuscript presented in an intelligible fashion and written in standard English?

Reviewer #1: Yes

Reviewer #2: Yes

Reviewer #3: Yes

5. Review Comments to the Author

Reviewer #1: i would like to start by appreciating all the work done by authors in presenting this research article. This appears to be a multinational survey study and lot of work has been put in collecting data and its presentation among patients, providers of different countries. Since it is ongoing, the data has not been fully presented yet. However, the objective of this data collection is not very clearly explained. it is not very clear from abstract how the data collection and analysis will improve the clinical and self management of patients of AF in the context of multimorbidity. While selecting subjects for survey, there was no specific inclusion criteria used, subjects appear to be included from patients, providers and caregivers of AF with multimorbidity. Also, I did not find any exclusion criteria in supplemental material. Rest of the work looks great!

Reviewer #2: The work put forth by the authors in compiling the results of their study in "Key Needs, Quality Performance Indicators and Outcomes for Patients with Atrial Fibrillation and Multimorbidity: The AFFIRMO study" is highly commendable and much appreciated. The study's primary objective is to identify the needs, quality performance indicators (QPIs), and outcomes relevant to patients, caregivers, and healthcare professionals to improve the care of patients with Atrial Fibrillation. This study is relevant to Cardiovascular medicine literature as it focuses primarily on providing holistic care of Atrial Fibrillation which represents a significant number of our patients. 

The major takeaways from the article were 

1. The study identifies key needs and key outcomes of elderly atrial fibrillation patients.

Overall, the manuscript is highly insightful and well-written; however, I would like the authors to throw some light on some of the points mentioned below.

1 . Healthcare providers who participated in the survey were mostly above age 70 about 73%. Is this accurate? This does not correlate with years of practice distribution

2 . Can the authors expand on how the difference in healthcare systems across countries might have affected the key needs and outcomes list?

Reviewer #3: The authors a very important study where they analyze the healthcare goals of atrial fibrillation patients, their care givers and health care providers. Withstanding the main limitation of the the final list being based on UK patients, The study is overall very well done.

I have the following questions/suggestions to the authors:

-How was living with assistance defined?

-Is there any data regarding the locations of these patients- whether they are urban or rural? It would be interesting to know this since healthcare views tend to vary in these locations.

-Given the patient population is predominantly white, the applicability of the current study on a global scale is somewhat limited. I would recommend authors address this in their limitation section.

6. PLOS authors have the option to publish the peer review history of their article (what does this mean?). If published, this will include your full peer review and any attached files.

Reviewer #1: No

Reviewer #2: **Yes: **Aniesh Bobba

Reviewer #3: **Yes: **Mukunthan Murthi

---

## [Author Response · Author response to Decision Letter 0]

22 Jul 2024

Dear Professor Vij

We thank the Reviewers for their constructive comments and for the opportunity to revise our manuscript entitled “Key Needs, Quality Performance Indicators and Outcomes for patients with Atrial Fibrillation and Multimorbidity: The AFFIRMO Study". We have addressed the comments made by the Editor and by the Reviewers and provided a point-by-point-response below. We have also uploaded a revised copy of the manuscript with tracked changes, together with a clean copy. We hope that the revised manuscript is now suitable for publication in PLoS ONE.

Yours sincerely,

Professor Deirdre Lane and Dr Donato G. Leo

on behalf of the co-authors

EDITOR’S COMMENTS:

Authors’ reply: Many thanks for your comment. We have now revised our manuscript as the template suggests. Tracked changes are active in the revised manuscript.

 [The AFFIRMO project has received funding from the European Union’s Horizon 2020 research and innovation programme under Grant agreement No 899871.]. 

Authors’ reply: Yes, the above-mentioned statement is correct: “The funders had no role in study design, data collection and analysis, decision to publish, or preparation of the manuscript."

Authors’ reply: “data cannot be publicly shared for ethical reasons, as data contain potentially identifying or sensitive patient information, but data sets can be made available upon request to the research team”.

Authors’ reply. We have checked the reference list and made some minor edits (e.g., added page numbers for some references which were missing). We have not changed the current references nor added additional references, and we have not cited any paper that has been retracted

REVIEW COMMENTS TO THE AUTHOR

REVIEWER #1: 

Reviewer #1:

I would like to start by appreciating all the work done by authors in presenting this research article. This appears to be a multinational survey study and lot of work has been put in collecting data and its presentation among patients, providers of different countries. Since it is ongoing, the data has not been fully presented yet. 

Authors’ reply: We thank you for your appreciation of our work. The survey and Delphi study presented in this manuscript were completed prior to our submission and the overall Work Package (of which this is a part) is also complete. 

However, the objective of this data collection is not very clearly explained. 

Authors’ reply: We have now edited and provided more details at the end of the Introduction section (page 5), which now reads as follows:

“The AFFIRMO sub-study reported here aimed to identify the key needs, QPIs, and outcomes relevant to patients with AF and multimorbidity, their caregivers, and healthcare professionals (HCP) involved in their clinical management. This aim was to understand the key aspects of care that need to be improved in the management of these patients and to inform the clinical trial of the AFFIRMO study, where an empowerment toolbox will be provided to patients and caregivers, enabling them to receive information about their health conditions.”

it is not very clear from abstract how the data collection and analysis will improve the clinical and self management of patients of AF in the context of multimorbidity.

Authors’ reply: We have now edited the Abstract (Page 2) to make it clearer how this study will improve the clinical and self-management of patients with AF in the context of multimorbidity. It now reads: 

“The final key needs, QPIs and outcomes obtained from the Delphi process will inform the AFFIRMO clinical trial, which aims to test the iABC app which incorporates an empowerment toolbox for patients and their caregivers, providing information to improve patient engagement and empowerment to help improve the clinical and self-management of patients with AF in the context of multimorbidity.”

While selecting subjects for survey, there was no specific inclusion criteria used, subjects appear to be included from patients, providers and caregivers of AF with multimorbidity. 

Authors’ reply: Eligible criteria were defined in the manuscript in the Methods section (Page 6), under the details of the online survey, as follows:

“The survey was open to patients with AF and one or more concomitant chronic health conditions (multimorbidity), and their caregivers (dyads and non-dyads), and to HCPs managing patients with AF and one or more concomitant chronic health conditions.”

Also, I did not find any exclusion criteria in supplemental material. Rest of the work looks great!

Authors’ reply: We have now added the exclusion criteria in the Methods section, page 6:

“Exclusion criteria were as follows: (i) inability to consent, (ii) moderate or severe cognitive impairment (e.g. dementia), (iii) presence of moderate or severe conditions that may prevent completion of the survey, (iv) inability to complete the survey online, and (v) unwilling to participate in the study.”

REVIEWER #2: 

Reviewer #2: The work put forth by the authors in compiling the results of their study in "Key Needs, Quality Performance Indicators and Outcomes for Patients with Atrial Fibrillation and Multimorbidity: The AFFIRMO study" is highly commendable and much appreciated. The study's primary objective is to identify the needs, quality performance indicators (QPIs), and outcomes relevant to patients, caregivers, and healthcare professionals to improve the care of patients with Atrial Fibrillation. This study is relevant to Cardiovascular medicine literature as it focuses primarily on providing holistic care of Atrial Fibrillation which represents a significant number of our patients.

The major takeaways from the article were 

1. The study identifies key needs and key outcomes of elderly atrial fibrillation patients.

Overall, the manuscript is highly insightful and well-written; however, I would like the authors to throw some light on some of the points mentioned below.

1. Healthcare providers who participated in the survey were mostly above age 70 about 73%. Is this accurate? This does not correlate with years of practice distribution

Authors’ reply: We think there may have been a misunderstanding with the data presented in our manuscript. The mean age you have highlighted relates to the patients that have participated in the online survey (Table 1, page 19), not to the healthcare provider (HCP) group. The mean age for the HCP group is 47.4 (SD±10.6) years, as reported in Table 3 (page 23) of the manuscript. This is also reported in the results of the online survey (page 8):

“A total of 1,305 participants completed the online survey. Of these, 659 (50.5%) were patients with a mean (SD) age of 70.9 (10.2) years and 348 (52.8%) female; 201 (%) were caregivers, mean (SD) age 58.3 (15.2) years and 147 (73.1%) female; and 445 (%) were HCPs, mean (SD) age 47.4 (10.6) and 257 (57.2%) female.”

2 . Can the authors expand on how the difference in healthcare systems across countries might have affected the key needs and outcomes list?

Authors’ reply: We have now expanded on this in the limitations (page 12), and it now reads as follows:

“The presence of only UK patient and caregiver participants in the Delphi process (due to the need of a common language for the discussion) is a limitation, which may have presented viewpoints related to one specific healthcare system, and may have influenced the key needs, QPIs, and outcomes selected. However, HCPs from five countries contributed to the Delphi process which ensured diversity in healthcare system perspectives. Further, the Delphi workshops discussed the needs, QPIs and outcomes identified by patients, caregivers and HCPs from five European healthcare systems via the on-line surveys, which minimises the possible UK-biased perspective of healthcare.”

REVIEWER #3: 

Reviewer #3: The authors a very important study where they analyze the healthcare goals of atrial fibrillation patients, their care givers and health care providers. Withstanding the main limitation of the final list being based on UK patients, The study is overall very well done.

I have the following questions/suggestions to the authors:

-How was living with assistance defined?

Authors’ reply: We did not provide any specific definition of ‘living with assistance’. This data was part of the self-reported information collected by the online survey. Therefore, patients identified themselves as living with assistance or not (as reported in Table 1, page 19). 

-Is there any data regarding the locations of these patients- whether they are urban or rural? It would be interesting to know this since healthcare views tend to vary in these locations.

Authors’ reply: No, unfortunately we did not collect further information on geographical location other than country.

-Given the patient population is predominantly white, the applicability of the current study on a global scale is somewhat limited. I would recommend authors address this in their limitation section.

Authors’ reply: We have now added this detail in the limitation section of our manuscript (page 12) which now reads as follows:

“Moreover, patients and caregivers who participated in the survey were mainly of White ethnicity which may reduce the generalisability of our findings.”

---

## [Editor Report · Decision Letter 1]

26 Aug 2024

Key Needs, Quality Performance Indicators and Outcomes for patients with Atrial Fibrillation and Multimorbidity: The AFFIRMO Study.

PONE-D-24-12600R1

Dear Dr. Lane,

We’re pleased to inform you that your manuscript has been judged scientifically suitable for publication and will be formally accepted for publication once it meets all outstanding technical requirements.

Kind regards,

Aviral Vij

Academic Editor

PLOS ONE
---

## [Editor Report · Acceptance letter]

30 Aug 2024

PONE-D-24-12600R1 

PLOS ONE

Dear Dr. Lane, 

I'm pleased to inform you that your manuscript has been deemed suitable for publication in PLOS ONE. Congratulations! Your manuscript is now being handed over to our production team.

Kind regards, 

on behalf of

Dr. Aviral Vij 

Academic Editor

PLOS ONE